# On Bitrates of Very Sparse Superposition Codes

## Abstract

*Sparse autoencoders* have been used to interpret activity inside large language models as "superposition codes" for sparse, high-dimensional signals. The encoder layers of these autoencoders use simple methods, which we will call "one-step estimates," to read latent sparse signals from vectors of hidden neuron activations. This work investigates the reliability of one-step estimates on a generic family of sparse inference problems. We show that these estimates are remarkably inefficient from the point of view of coding theory: even in a "very sparse" regime, they are only reliable when the dimension of the code exceeds the entropy of the latent signal by a factor of 2.7 dimensions per bit. In comparison, a very naive iterative method called matching pursuit can read superposition codes given just 1.3 dimensions per bit. This opens the question of whether neural networks can achieve similar bitrates in their internal representations.

## 1. Introduction

If each neuron in a neural network signaled a meaningful "feature" of its input, we could hope to reverse-engineer the network's overall behavior on a neuron-by-neuron basis. However, individual neurons of real-world networks often lack clear interpretations. For example, both language models and vision models have been found to learn neurons that correlate simultaneously with apparently unrelated features. (See for example (Nguyen et al., 2016), (Zhang & Wang, 2023) and (Olah et al., 2020).)

The difficulty of interpreting a network in terms of its local activity—and in particular, the appearance of so-called "polysemantic neurons"—is not surprising from a connectionist viewpoint. Since at least the 1980s, proponents of neural networks have argued that these systems

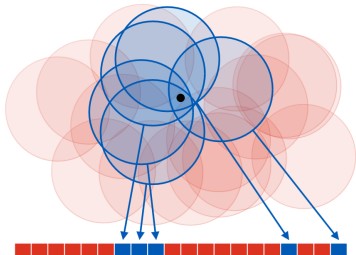

Figure 1. A coarse code representing a point on a plane. Each "neuron," drawn as a red or blue square, encodes whether the point belongs to an associated "receptive field." Although no neuron gives specific information on the position of the point, the overall code determines its position with reasonable accuracy.

may naturally use **distributed representations**—coding schemes where individual features are represented by patterns spread over many neurons, and conversely where each neuron carries information on many features. (This term was apparently coined in (Rumelhart et al., 1986), Chapter 3.) In contrast, a *local* representation would dedicate each neuron to a single feature. (See (Thorpe, 1989) for a general discussion of local and distributed codes.) Figure 1 illustrates a classic example of a coarse code, one kind of distributed representation.

It is not clear how deep neural networks learn to represent information in their hidden layers or to what extent this information can be interpreted. However, should "interpretable features" exist, the connectivist viewpoint makes it natural that they would be stored with non-local codes. This is a common assumption in interpretability research today; for example, when (Meng et al., 2022) intervened on an MLP layer of a language model to "edit" a factual association, both the "subject" and the "fact" were modeled as vectors of neuron activations rather than as individual neurons.

How can we infer latent features learned by a neural network? One simple proposal is to model an activation vector $x$ as a linear projection

$$x = Fy$$

of some high-dimensional and *sparse* vector $y$ of latent fea-

[1]Anonymous Institution, Anonymous City, Anonymous Region, Anonymous Country. Correspondence to: Anonymous Author <anon.email@domain.com>.

Preliminary work. Under review by the International Conference on Machine Learning (ICML). Do not distribute.

tures. We refer to the columns of $F$ as codewords and the whole matrix $F$ as a dictionary. Since $x$ is a linear superposition of codewords, we will call it a **superposition code** for $y$. The task of inferring the sparse vector $y$ from $x$ is known as sparse reconstruction, and the task of inferring the dictionary $F$ from a distribution over $x$ is called dictionary learning. Both of these problems have been studied in the field of compressive sensing, although with different applications in mind. (See (Elad, 2010) for a review of classic work in the context of signal and image processing.)

Already in 2015, (Faruqui et al., 2015) used a dictionary learning method to derive sparse latent codes for word embeddings and argued that these latents were more interpretable than the original embedding dimensions. More recently, a series of works beginning with (Yun et al., 2021) have applied dictionary learning to the internal representations of transformer-based language models. (Cunningham et al., 2023) suggested the use of **sparse autoencoders** (SAEs) and (Templeton et al., 2024; Gao et al., 2024) scaled sparse autoencoders to production-size large language models.

Sparse latents learned by SAEs are often highly intepretable, and (Templeton et al., 2024) showed that intervening at the level of features allows "steering" language models in predictable ways. However, even SAEs with very high-dimensional latents suffer from an apparently irreducible reconstruction error (Gao et al., 2024). Understanding the limitations of SAEs—and dictionary learning in general—is an important open question in interpretability (Sharkey et al., 2025).

## 2. Contributions

To infer a latent representation $y$ from an activation vector $x$, sparse autoencoders use an estimate like $\hat{y}(x) = \sigma(Gx)$ for some learnable matrix $G\colon \mathbb{R}^{N \times d}$ and some simple non-linear thresholding function $\sigma$. Meanwhile, the literature on compressive sensing is concerned mainly with *iterative* methods for sparse inference. Throughout this paper, we will refer to autoencoder estimates as "one-step estimates."

It is natural that iterative methods for sparse reconstruction will perform more reliably than one-step estimates, but the nature of this gap is not obvious in general. Informally speaking, how bad are one-step estimates?

In this work, we answer this question in a toy scenario designed to model the "very sparse" latents learned by sparse autoencoders in practice. Our main contributions are the following.

1. We prove a theoretical guarantee on the performance of one-step methods and indicate simple "rules of thumb" that hold in practice. (See Section 3.3.)

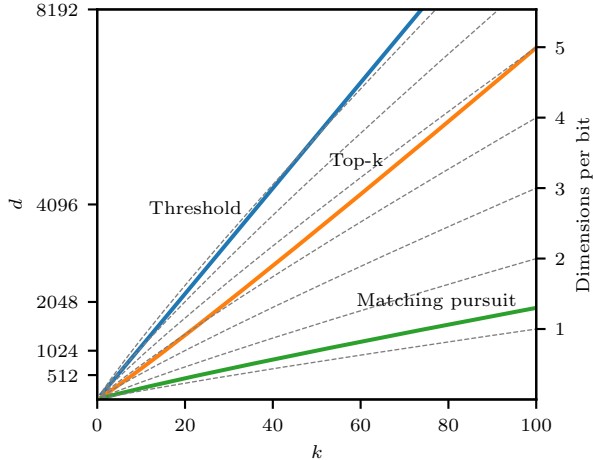

Figure 2. An overview of the minimum codeword dimension $d$ required for three different methods to reliably decode a uniformly chosen $k$-sparse subset of $\{1, \ldots, 2^{20}\}$ from a superposition of Rademacher codewords. Threshold and top-$k$ decoding are "one-step" methods used by sparse autoencoders, while matching pursuit is a simple iterative method. The inverse "bitrate" $d/H(k)$, where $H(k) = \log_2 \binom{N}{k} \approx k \log_2(eN/k)$, is indicated by the right axis.

2. We show empirically that the gap between one-step methods and iterative methods is significant, *even for very sparse latents*. In comparison to a simple method called matching pursuit, one-step methods require the dimension $d$ of the superposition code to be larger by a constant factor. (See Section 3.5.)

From the point of coding theory, one natural measure for the efficiency of a sparse recovery method is its *bitrate*: that is, the ratio $H/d$ between the entropy $H$ of the latent signal and the minimum dimension $d$ of the code $x = Fy$ needed to recover $y$. In this language, matching pursuit can decode "very sparse" superposition codes at a rate of roughly one bit per dimension. On the other hand, one-step methods require upwards of 2.7 dimensions per bit. This rate increases as $y$ becomes less sparse; for a latent vector $y \in \mathbb{R}^{2^{20}}$ with 100 non-zero entries, one-step estimates require about *5 dimensions per bit*. (See Figure 2.)

How "efficient," in terms of bitrate, are the codes used by real neural networks? On one hand, it would not make sense for a network to use a code that requires a lengthy iterative decoding process before it can be used. On the other hand, it may still be possible for a network to learn to use codes that are "too efficient" to be entirely decoded by a one-step estimate. Overall, we hope this question informs future work on modeling distributed representations.

## 3. Encoding Sets with Superposition Codes

We begin by describing the "toy scenario" to be studied.

Given a large number $N$, consider a map $F$ that "encodes" each subset $y \subseteq [N] = \{1, \ldots, N\}$ by a linear combination

$$x = Fy = \sum_{i \in Y} f_i \in \mathbb{R}^d,$$

where the vectors $\{f_i \in \mathbb{R}^d : i \in [N]\}$ are chosen in advance and where the dimension $d$ of the encoding is expected to be much smaller than $N$. As above, we call the vectors $f_i$ codewords for the elements of $[N]$ and call the image $Fy$ a superposition code for the set $y$. It will often be useful to view $y$ as a vector in $\{0, 1\}^N$ with coefficients

$$y_i = \begin{cases} 1 : i \in y \\ 0 : \text{otherwise} \end{cases}$$

and view $F$ as a matrix of column vectors $[f_1 \ \ldots \ f_N]$, called the dictionary. For simplicity, we'll model our subset as a random variable $Y$ uniformly distributed over the subsets of some fixed size $k \ll N$.

Ultimately, we are interested in understanding what might limit the success of SAEs and how other sparse dictionary learning methods may be designed. As a first step, this work addresses the following question.

**Question 1.** *When can $Y$ be reliably decoded from the superposition code $X = FY$ with the methods used by sparse autoencoders? Can other computationally efficient methods do significantly better?*

Specifically, we're interested in understanding how large the dimension $d$ needs to be as a function of $(N, k)$ for some class of method to recover $Y$, assuming the dictionary $F$ is known. (We do not study the problem of learning the dictionary.) Since $Y$ is a discrete variable, we will focus on conditions for *exact* recovery. We'll also focus on a regime where $Y$ resembles the very sparse latent representations learned by sparse autoencoders trained on large language models. (Gao et al., 2024) discusses scaling the number of latent features on the order of $N = 2^{20}$ with sparsity on the order of $k = 2^8$, so we use this as our reference.

To map vectors of activations to latent sparse representations—in our language, to infer $X$ from $Y$—sparse autoencoders essentially employ one-layer networks. For example, (Templeton et al., 2024) used a ReLU unit to estimate each coefficient of $Y$. Since the coefficients $Y_i$ in our toy scenario are either $0$ or $1$, a natural analog would be a thresholding rule of the form

$$\hat{Y}_i(x) = \begin{cases} 1 : \langle \lambda_i, X \rangle \geq 1 \\ 0 : \text{otherwise} \end{cases}$$

Since the number $k$ of non-zero coefficients is known beforehand, we can also choose the threshold adaptively so that only $k$ of the $\hat{Y}_i$ are non-zero. This is called top-$k$ decoding. (Gao et al., 2024) showed that, in practice, top-$k$ autoencoders perform better than their ReLU variants. We refer to both approaches as "one-step estimates."

On the other hand, the field of compressive sensing offers a vast literature on *iterative* methods to recover a sparse vector from a linear projection. It is known that, in general, iterative methods are much more reliable than one-step estimates. Indeed, the *first iteration* of an iterative shrinkage method (see Chapter 6 of (Elad, 2010)) is formally identical to the kind of ReLU network employed by (Templeton et al., 2024). However, to our knowledge, a comparison of one-step estimates with iterative methods in the very sparse regime encountered by sparse autoencoders has so far been lacking.

The following sections are organized as follows.

- Section 3.1 reviews some basic ideas from information theory and introduces bitrate as a measurement for the efficiency of an inference method.

- Section 3.2 reviews the idea of a matched filter and motivates the two one-step estimates we will consider.

- Section 3.3 studies the reliability of one-step estimates when the dictionary $F$ is random.

- Section 3.4 argues that random dictionaries are "almost optimal" when $k \ll N$.

- Section 3.5 discusses the empirical performance of an iterative method called matching pursuit.

### 3.1. Information Theory Bounds

In practice, each dimension of the superposition code $FY$ carries a finite amount of information on the set $Y$. At best, the information that one dimension can store is determined by the number of states in its numeric datatype—a 16 bit floating point can store nearly 16 bits, and so on. However, under the moderate assumption that the projection $FX$ can still be decoded after the addition of a certain level of white noise, classic results from information theory put more realistic bounds on the dimension of our encoding.

**Proposition 1.** *For a given dictionary $F \in \mathbb{R}^{d \times N}$, suppose there exists a decoding map $D$ so that*

$$D(FY + Z) = Y$$

*with probability at least $(1-p)$, where $Z$ is a vector of i.i.d. Gaussians with variance $V_Z$. Suppose additionally that the*

*maximum variance of any coefficient of the code $X = FY$ if $V_X$. Define*

$$C = \frac{1}{2} \ln \left( 1 + \frac{V_X}{V_Z} \right).$$

*Then*

$$d \geq C^{-1} \left( (1-p) \ln \binom{N}{k} - \ln 2 \right).$$

(See Appendix A for a standard proof.) When $p$ is small and $\ln \binom{N}{k}$ is large, this means roughly that the "bitrate"

$$R = \log_2 \binom{N}{k} \bigg/ d$$

cannot exceed the "channel capacity" $C/\ln 2$. (We alternate between bits and nats as convenient.) On the other hand, a classic result of information theory is that, as some block size parameter goes to infinity, there exist arbitrarily reliable coding schemes that essentially meet the channel capacity. In the remainder of this work, we will measure the minimum dimension $d$ required for a certain inference method to recover $Y$ in terms of the corresponding bitrate $R$.

Of course, it will be useful to have an estimate for the entropy $\ln \binom{N}{k}$. For fixed $k$, $\binom{N}{k}$ is a polynomial in $N$ and so

$$\ln \binom{N}{k} = k \ln N + O(1).$$

Indeed, $k \ln N$ is the entropy of an array of $k$ elements of $[N]$ drawn with replacement. A better estimate for the entropy of a small subset is

$$\ln \binom{N}{k} \leq k \ln(eN/k) = k \ln N - k \ln k + k.$$

In fact, when $k$ is small compared to $N$, $k \ln(eN/k)$ is an extremely good approximation. For example,

$$\ln \binom{2^{20}}{2^8} \approx 2^8 \ln(2^{20} e/2^8) = 128(1 + 12 \ln 2)$$

holds with a relative error of about $0.3\%$. (See Appendix B for a discussion of this estimate.)

### 3.2. Matched Filters and One-Step Estimates

Now, we turn to the problem of decoding a superposition code. Let's begin by reviewing the simpler problem of inferring a random scalar $S$ from a sum

$$X = Sf + Z \qquad (1)$$

where $f \in \mathbb{R}^d$ is known but the "noise term" $Z \in \mathbb{R}^d$ is an unobserved Gaussian vector. In signal processing, the problem of recovering an unobserved variable from a noisy process is known as *filtering*.

In a linear system with Gaussian noise, like Equation (1), optimal filtering can be done using a linear function of the measurement data. Specifically, suppose $Z$ has mean zero and non-singular covariance $\Sigma$, and define an inner product by $\langle v, w \rangle_\Sigma = x^T \Sigma^{-1} y$. Then the posterior of $S$ conditional on $X$ is determined by the function

$$\lambda(X) = \frac{\langle f, X \rangle_\Sigma}{\|f\|_\Sigma^2},$$

which we will call the **matched filter** for $S$. If $S \in \{0, 1\}$ is a binary variable, a routine calculations shows that the log odds of the posterior on $S$ is given by

$$\begin{aligned}
&\ln \frac{\mathrm{P}(S = 1 | X = x)}{\mathrm{P}(S = 0 | X = x)} \\
&= \rho \left( \lambda(x) - \frac{1}{2} \right) + \ln \frac{\mathrm{P}(S = 1)}{\mathrm{P}(S = 0)}, \qquad (2)
\end{aligned}$$

where $\rho = \|f\|_\Sigma^2$ is the "signal-to-noise ratio" of the filter $\lambda$. See Appendix D for a review.

We now return to our original problem. Let's focus on estimating just one scalar $Y_i$ from the sum

$$X = Y_i f_i + \sum_{j \neq i} Y_j f_j.$$

The "noise term" here is not Gaussian, and the exact Bayesian posterior on $Y_i$ turns out to be intractable in general. However, we can try to estimate $Y_i$ by approximating $\sum_{j \neq i} Y_j f_i$ by a Gaussian vector of the same covariance. The corresponding matched filter for $Y_i$ can be understood as a kind of least squares estimate.

In the following, let us assume that the codewords $f_i \in \mathbb{R}^d$ are unit vectors. (It is natural for all the codewords $f_i$ to have the same magnitude if each coefficient $Y_i$ needs to be encoded with the same precision, as they do in our scenario.) If we assume further that the empirical distribution over codewords $f_i$ is approximately isotropic, then the matched filter for $Y_i$ is approximately

$$\lambda_i(X) = \langle f_i, X \rangle.$$

(If the distribution over codewords is not isotropic, we can first apply a linear transformation to "whiten" the distribution of $X$.)

A **one-step estimate** is an estimate for $Y$ that relies directly on the matched filters $\lambda_i$. From Equation (2), the maximum likelihood estimate for $Y_i$ under our simplified Gaussian model is 1 if

$$\langle f_i, X \rangle \geq \frac{1}{\rho} \ln \frac{\mathrm{P}(Y_i = 1)}{\mathrm{P}(Y_i = 0)} + \frac{1}{2}$$

and 0 otherwise. If we assume the signal-to-noise ratio $\rho$ is very large, the decision boundary becomes approximately $1/2$. This leads to the simpler of the two one-step estimates that we will consider.

**Definition 1.** *Given $X = FY$, the **threshold decoding** is*

$$\hat{Y}_i = \begin{cases} 1 : \langle f_i, X \rangle \geq 1/2 \\ 0 : otherwise. \end{cases}$$

On the other hand, if we know (or guess) the size $k$ of the set $Y$ in advance, the following is a natural way to make use of that information. (In the context of sparse autoencoders, this method was introduced by (Makhzani & Frey, 2014).)

**Definition 2.** *Given $X = FY$, the **top-$k$ decoding** is the set $\hat{Y}$ of $k$ elements whose codewords $f_i$ have largest inner products with $X$. (Ties are broken arbitrarily.)*

Note that whenever threshold decoding succeeds at recovering $Y$, top-$k$ decoding succeeds as well.

### 3.3. One-Step Estimates with Random Codewords

In this section, we show rigorously that one-step estimates are reliable so long as $d = \Omega(k \ln N)$ and the dictionary $F$ is random. Our theoretical results agree with numerical experiments, and we find that remarkably simple "rules of thumb" govern the performance of one-step estimates in practice. (See Figure 3.)

If inner products $\langle f_i, f_j \rangle$ between distinct codewords are "small enough" in some sense, then the matched filters $\langle f_i, X \rangle$ will be reliable and we can expect one-step estimates to succeed. Indeed,

$$\langle f_i, X \rangle = \left\langle f_i, \sum_j Y_j f_j \right\rangle = \sum_j Y_j \langle f_i, f_j \rangle$$
$$= Y_i + \underbrace{\sum_{j \neq i} Y_j \langle f_i, f_j \rangle}_{\xi_i}, \tag{3}$$

where the total "crosstalk" $\xi_i$ is a sum of either $(k-1)$ or $k$ inner products $\langle f_i, f_j \rangle$.

One simple way to produce a dictionary of almost-orthogonal codewords is to choose them randomly. For example, the following fact is representative of many similar results in high-dimensional geometry.

**Proposition 2.** *Let $d > 2\epsilon^{-2}(2 \ln N + \ln p^{-1})$, and let*

$$\{F_1, \ldots, F_N\} \subseteq \{-1/\sqrt{d}, 1/\sqrt{d}\}^d$$

*be random vectors with independent, uniformly distributed entries. Then $|\langle F_i, F_j \rangle| < \epsilon$ for all $i \neq j$ with probability at least $(1-p)$.*

See Appendix C for a review.

Let's call a pair $(v, w)$ of vectors "$\epsilon$-orthogonal" when $|\langle v, w \rangle| < \epsilon$. When all codewords are pairwise $\epsilon$-orthogonal in the sense of Proposition 2, the crosstalk $\xi_i$ in Equation (3) is bounded strictly by $\epsilon k$ in absolute value. Putting $\epsilon = k/2$ gives the following corollary.

**Corollary 1.** *Let $d \geq 8k^2(2 \ln N + \ln p^{-1})$, and let $F \in \mathbb{R}^{d \times N}$ be a dictionary of random codewords in the conditions of Proposition 2. Then with probability at least $(1-p)$, every $k$-element subset $Y \subseteq [N]$ is recovered from its superposition code $FY$ by threshold decoding.*

For fixed $k$, we conclude that the dimension $d$ of our codewords only needs to grow as $\Omega(\ln N)$. However, the factor of $16k^2$ turns out to be very pessimistic; in practice, for *almost all* sets to be reliably encoded, we only need $\Omega(k \ln N)$ dimensions.

**Proposition 3.** *Let $F \in \mathbb{R}^{d \times N}$ be a Rademacher dictionary in the conditions above. Fix a $k$-element set $y \in [N]$ and some $p \in (0, 1)$. If*

$$d \geq 8k(\ln N + \ln p^{-1}),$$

*then $y$ is accurately recovered from the random variable $X = Fy$ by threshold decoding with probability at least $(1-p)$.*

As a heuristic guide for this result, consider the crosstalk $\xi_i$ encountered by a matched filter $\langle f_i, X \rangle$. If we view the other $(N-1)$ codewords as random Rademacher vectors $F_j$, we find that each inner product $\langle f_i, F_j \rangle$ is a sum of $d$ independent Rademacher variables scaled to have total variance $1/d$. It follows that the variance of $\xi_i$ is at most $k/d$. To keep the power of this crosstalk below some fixed threshold, we conclude that $d$ must grow linearly with respect to $k$. For a full proof, see Appendix E.

Note that, unlike Corollary 1, Proposition 3 does not guarantee that any *fixed* dictionary can reliably encode many sets $y$. However, we can easily derive such a guarantee with a Markov inequality.

**Corollary 2.** *Let $F \in \mathbb{R}^{d \times N}$ be a Rademacher dictionary as above and let $\epsilon, p > 0$. If*

$$d \geq 8k(\ln N + \ln(\epsilon p)^{-1}),$$

*then with probability at least $(1-p)$ it is true that at least $(1-\epsilon)\binom{N}{k}$ subsets $y$ are accurately decoded from their images $X = Fy$ by threshold decoding.*

The prediction of Proposition 3 agrees well with numerical experiments, graphed in Figure 3. In fact, even as $N$ varies over several orders of magnitude, the slightly weaker condition $d \geq 8k \ln N$ characterizes the regime where the

set $Y$ can be decoded with reasonably high probability by threshold decoding.

Top-$k$ decoding performs significantly better but admits a similar "rule of thumb": for all values of $N$ trialed,

$$d = 4k \ln kN$$

is very close to the smallest dimension needed for top-$k$ decoding to succeed with high probability. See Appendix F for an informal derivation of this bound.

### 3.4. Are Random Codewords Optimal?

So far, we've considered the performance of threshold and top-$k$ decodings at recovering a subset from a superposition code with a *random* dictionary $F$. One natural question is whether we can do better if the dictionary is optimized. Of course, when $d \geq N$, we can make the codewords $f_i$ exactly orthogonal. For this reason, the performance of one-step decodings shown in the top row of Figure 3 is much worse than is possible; we never need more than $N$ dimensions to store a latent vector of dimension $N$.

However, when the ratio $d/N$ is small—say, smaller than $1/10$—we conjecture that optimizing the dictionary gives practically no improvement over a random initialization. Unfortunately, we are not aware of a theoretical justification for this fact.

To see why this may be true, recall the "crosstalk" terms

$$\xi_i = \sum_{j \neq i} Y_j \langle f_i, f_j \rangle$$

from the previous section. For each $i$, this is a sum of between $k$ and $(k-1)$ numbers drawn without replacement from the sequence

$$(\langle f_i, f_j \rangle)_{j \neq i}.$$

Let's fix the dictionary $F$ and consider the empirical distribution defined by this sequence. Suppose this distribution has zero mean and variance

$$\gamma_i(F) = \frac{1}{N-1} \sum_{j \neq i} \langle f_i, f_j \rangle^2.$$

When $k$ is moderately large but much smaller than $N$, we expect the crosstalk $\xi_i$ to behave like a centered Gaussian with variance $k\gamma_i$. Specifically, we expect that the probability of its tail events with respect to the random set $Y$ will be governed by the product $k\gamma_i$. If we assume that tail events for the different variables $\xi_i$ are "sufficiently independent," we conclude overall that the typical value of $\gamma_i(F)$ is the limiting factor for the reliability of one-step estimates.

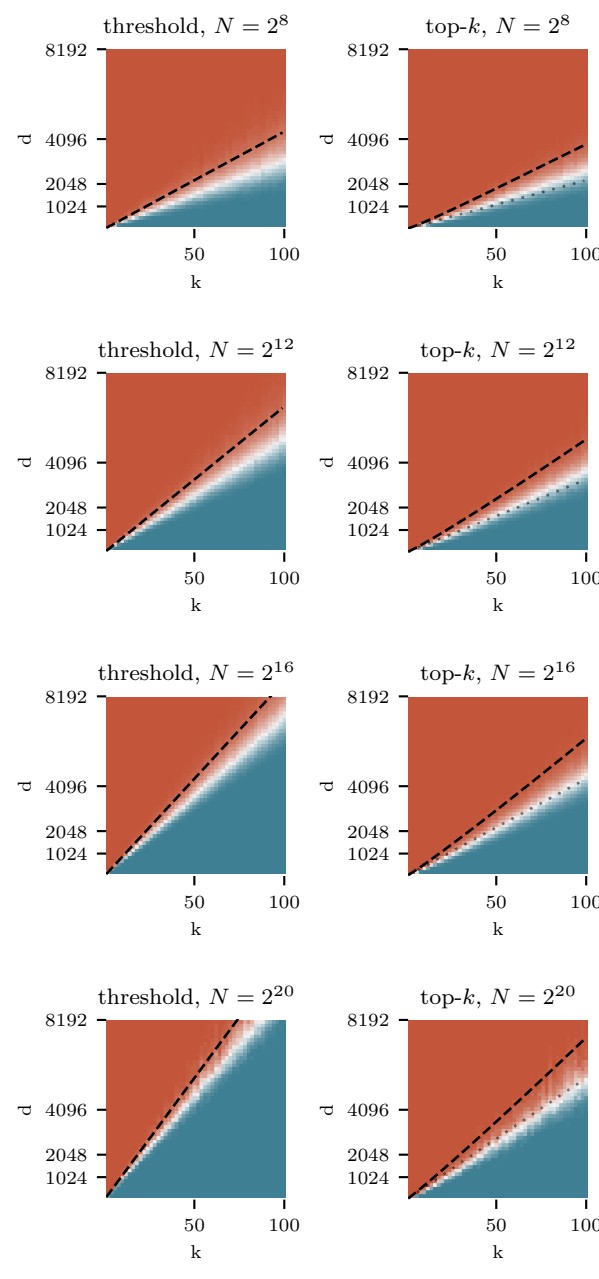

Figure 3. Empirical performance of threshold decoding (left) and top-$k$ decoding (right) at the problem of recovering a $k$-element subset of $[N]$ from a projection into $d$ dimensions by a Rademacher random matrix. In the left column, we plot the relation $d = 8k \ln N$. On the right, we plot $d = 4k \ln(kN)$ and its lower bound of $d = 4k \ln N$.

A dictionary chosen to have smaller "interference scales" $\gamma_i$ would, in particular, have smaller average squared interference

$$\gamma(F) = \frac{1}{N} \sum_{i=1}^{N} \gamma_i = \binom{n}{2}^{-1} \sum_{i \neq j} \langle f_i, f_j \rangle^2.$$

For a random dictionary $F$, $\gamma(F)$ equals $1/d$ in expectation. Can we decrease this value significantly by optimization?

Using projected gradient descent, we minimized $\gamma(F)$ subject to the constraint of maintaining unit norm codewords. We tested dictionaries with between $N = 64$ and $65536$ codewords and with codeword dimensions between $d = 16$ and $1024$. In each case, we initialized with a random Rademacher dictionary and optimized to convergence with standard criteria. Our results are plotted in Figure 4.

As $d$ approaches $N$, we find that the optimal value $\gamma_{\text{opt}}$ of $\gamma(F)$ converges to 0, as expected. On the other hand, when $d \ll N$, $\gamma_{\text{opt}}$ is very close to $1/d$, its expected value under a random initialization. For example, with $N = 2^{16}$ (not plotted), the optimal value of $\gamma(F)$ is indistinguishable from $1/d$ on a log-log plot.

Furthermore, we find a striking regularity. Empirically, the ratio $\gamma_{\text{opt}}/d^{-1} = d\gamma_{\text{opt}}$ between the optimal value of $\gamma$ and its expected value at initialization turns out to be a function of the relative dimension $d/N$. Since this holds as $N$ ranges over several orders of magnitude, it is natural to believe it may hold in general.

**Claim 1.** *For given $(N, d)$, the optimal value of $\gamma(F)$ for a dictionary $F \in \mathbb{R}^{d \times N}$ of unit norm codewords is*

$$\gamma_{opt}(N, d) = \frac{\kappa(d/N)}{d}$$

*for some function $\kappa$. Furthermore, $\kappa(r)$ is close to 1 for small values of $r$.*

If true, this means that the values $\gamma_i(F)$ governing the scale of crosstalk suffered by matched filters can't be made significantly smaller than $1/d$ when $d \leq \epsilon N$ for small $\epsilon$.

We're not aware of theoretical results in this direction. Note in particular that this is not obviously related to work on sphere packing (see (Cohn & Zhao, 2014)) since we are interested in the *scale* of the distribution of inner products rather than in maximum values.

### 3.5. Comparison with Compressive Sensing

Together, Section 3.3 and Section 3.4 provide strong evidence that when $k \leq \epsilon N$, one-step estimates need nearly $d \geq 4k \ln N$ dimensions to read a subset from a superposition code, even when the dictionary $F$ is optimized. In the sense of Section 3.1, this means that the "bitrate" of a

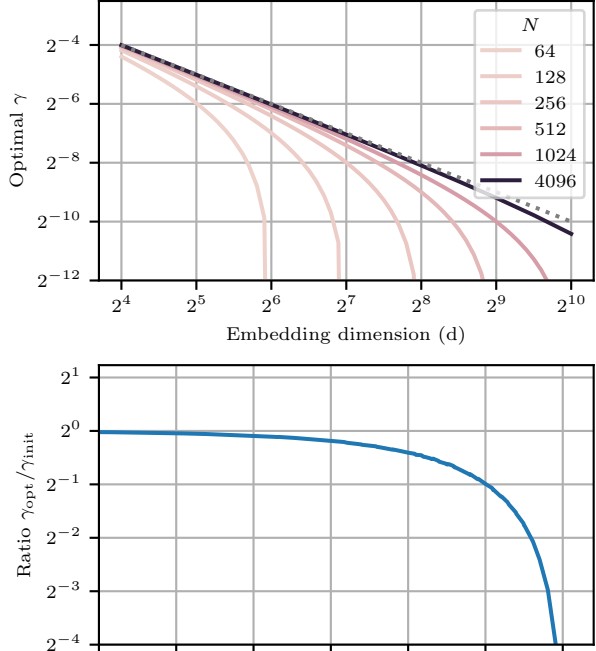

Figure 4. **Top**: The mean squared interference $\gamma(F)$ of a dictionary $F$ obtained by running projected gradient descent to convergence. The dotted line shows $\gamma_{\text{init}} = 1/d$, the mean squared interference attained in expectation by a random initialization. The best interference for $N = 2^{16}$ found by gradient descent (not graphed) is nearly indistinguishable from the dotted line. **Bottom**: A plot of ratio $\gamma_{\text{opt}}/\gamma_{\text{init}}$ by which gradient descent improves $\gamma$ relative to its expected value at initialization against the ratio $d/N$ between codeword dimension and dictionary size.

superposition code is at most

$$R = \log_2 \binom{N}{k} \Big/ (4k \ln N)$$
$$\leq \frac{k \log_2 (eN/k)}{4k \ln N} = \frac{1}{4 \ln 2} \left( 1 - \frac{\ln k - 1}{\ln N} \right) \quad (4)$$

bits per dimension. (Note that $4 \ln 2 > 2.7$.)

There are several ways to interpret this conclusion. On one hand, it means that one-step estimates are "asymptotically inefficient" in terms of required bitrate when $k$ is moderately large compared to $N$. More specifically, in a regime where $N$ goes to infinity but $\ln k / \ln N$ converges to 1, we predict that one-step estimates only succeed when the bitrate $R$ converges to zero.

In particular, one-step estimates are asymptotically inefficient when $k/N \geq \epsilon$ for some positive $\epsilon$. Indeed, to have $d \geq 4k \ln N$ we would need $d = \Omega(N \ln(N))$, while the

entropy of $Y$ grows no faster than $O(N)$. On the other hand, a hallmark result of compressive sensing implies that, when $k/N \leq \epsilon$, the vector $y$ can be recovered from its image $Fy$ under a random projection by a certain *convex* optimization problem so long as $d \geq \kappa(\epsilon)N$ for some constant $\kappa(\epsilon)$; for example, see (Candes & Tao, 2005). The failure of our one-step estimates in this particular regime is easy to prove.

On the other hand, in a sparser regime where $\ln k / \ln N < \epsilon$ for some $\epsilon < 1$, it follows from Proposition 3 that one-step estimates are "information-efficient" in the sense that they can be decoded from superposition codes with bitrate larger than some positive $\delta$. However, it is also of interest to have *non-asymptotic* information on the required bitrate. From Equation (4) we find that one-step estimates need at least 2.7 bits per dimension. Can iterative methods do better?

There is an extensive literature on theory of compressive sensing. (Reeves et al., 2019) shows that, in our language, superposition codes with a random dictionary are essentially *optimal* in the information-theoretic sense when ideal maximum-likelihood inference is used as the decoder. A series of earlier works (Joseph & Barron, 2012; 2014; Rush et al., 2017) on superposition codes also showed that, under some conditions on $y$, certain decoding schemes admit bitrates up to theoretical channel capacity in the presence of Gaussian noise. However, to our knowledge, practical guarantees on the performance of iterative methods are not available for our range of $k$ and $N$.

Figure 5 shows the results of a numerical experiment using an iterative method called *matching pursuit*, first suggested in (Bergeaud & Mallat, 1995). This is a simple "greedy" algorithm that initializes $y = 0$ and, at each of $k$ iterations, increments the index of $y$ whose corresponding codeword has largest inner product with $x - Fy$.

We find empirically that matching pursuit far outperforms top-$k$ decoding for the range of $N$ and $k$ considered earlier. Remarkably, decoding tends to be successful with even odds when $d = k \log_2(eN/k)$, meaning that matching pursuit requires only slightly more than *one dimension per bit*. When $d \geq 1.3 \log_2(eN/k)$, decoding is very reliable when $N = 2^{16}$ and $N = 2^{20}$.

## 4. Conclusions and Future Work

Previous work showed that sparse autoencoders can help learn interpretable representations of the activity inside a neural network. However, the success of these methods is limited for reasons that are not yet well understood.

In this work, we have identified one point of view that might explain their limited success. In a toy scenario,

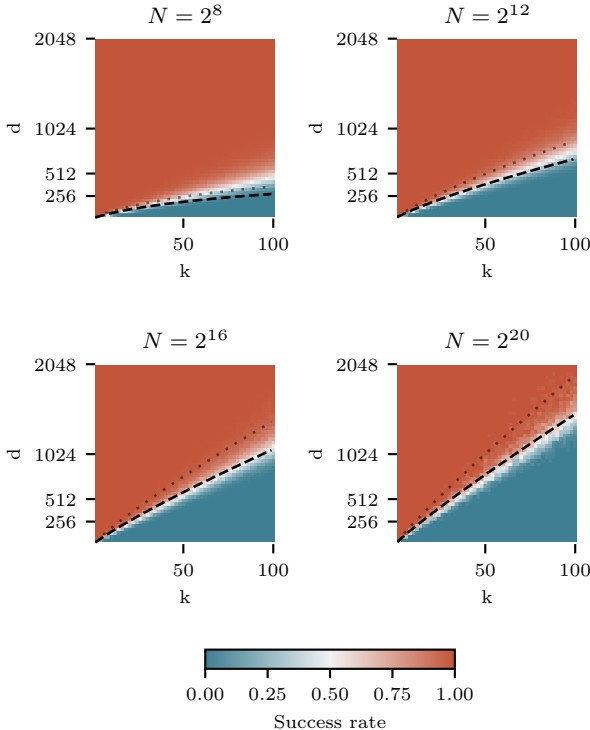

Figure 5. Empirical performance of matching pursuit at the problem of decoding a $k$-element subset of $[N]$ from a superposition code with Rademacher dictionary. Note the difference of vertical axis scale compared to Figure 3. The bold line shows the relation $d = k \log_2(eN/k)$, and the dotted line shows $d = 1.3k \log_2(eN/k)$.

we showed that the simple estimates these models use to infer sparse representations are less "efficient," in an information-theoretic sense, than a simple iterative method. This is true even when the signal to be inferred is extremely sparse. To our knowledge, this kind of explicit, non-asymptotic comparison was not previously available in the literature.

Of course, we do not suggest that the latent signal stored by a typical neural representation is well-modeled as a uniformly random $k$-sparse subset. However, the "bitrate gap" between one-step estimates and matching pursuit opens a natural question: how much information can neural networks typically encode in their internal activity? Can they, like matching pursuit, read around one bit of mutual information from each neuron? If they can, our findings suggest that sparse autoencoders may be fundamentally unable to decode their representations. Overall, we hope the point of view of coding efficiency helps inspire better interpretability methods in future work.

## 5. Impact Statement

This paper considers basic problems that may be relevant to interpretability of neural networks. We do not feel that any broader societal consequences need to be highlighted.

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

## A. Proof of Proposition 1

We restate Proposition 1 for convenience.

**Proposition.** *For a given dictionary $F \in \mathbb{R}^{d \times N}$, suppose there exists a decoding map $D$ so that*

$$D(FY + Z) = Y$$

*with probability at least $(1 - p)$, where $Z$ a vector of i.i.d. Gaussians with variance $V_Z$. Suppose additionally that the maximum variance of any coefficient of the code $X = FY$ if $V_X$. Define*

$$C = \frac{1}{2} \ln \left( 1 + \frac{V_X}{V_Z} \right).$$

*Then*

$$d \geq C^{-1} \left( (1 - p) \ln \binom{N}{k} - \ln 2 \right).$$

*Proof.* By results on the capacity of Gaussian channels (see (Thomas & Joy, 2006), Chapter 9) we can bound the mutual information between $X$ and $X + Z$ as

$$I(X, X + Z) \leq \frac{d}{2} \ln(1 + \rho)$$

where $\rho$ is an upper bound for the signal-to-noise ratio of each entry of $X + Z$. In our case, we can put $\rho = V_X / V_Z$.

Now, let $D$ is a decoding in the conditions above. Then a relaxation of Fano's inequality shows

$$I(Y, D(FY + Z)) \geq (1 - p) \ln \binom{N}{k} - \ln 2.$$

But since $I(Y, FY + Z) \geq I(Y, D(FY + Z))$, we conclude that overall

$$\frac{d}{2} \ln \left( 1 + \frac{V_X}{V_Z} \right) \geq (1 - p) \ln \binom{N}{k} - \ln 2.$$

$\square$

## B. Estimates for the Binomial Coefficient

To estimate $\ln \binom{N}{k}$, it is helpful to first remember the elementary inequalities

$$\left( \frac{N}{k} \right)^k \leq \binom{N}{k} \leq \left( \frac{eN}{k} \right)^k.$$

Taking logarithms gives

$$k \ln(N/k) \leq \ln \binom{N}{k} \leq k \ln(eN/k),$$

and so $\ln \binom{N}{k} = k \ln(N/k) + O(k)$.

In this work, we claimed the upper bound $k \ln(eN/k)$ is a very good approximation when $k \ll N$. To see why, substitute the leading-order Stirling approximation $\ln n! = n \ln n - n + O(\ln n)$ into the binomial coefficient to obtain

$$\ln \binom{N}{k} = (N - k) \ln \left( \frac{N}{N - k} \right) + k \ln \left( \frac{N}{k} \right) + O(\ln N).$$

Putting $s = k/N$, this simplifies to:

$$\ln \binom{N}{k} = h(s)N + O(\ln N),$$

where

$$h(s) = -s \ln s - (1 - s) \ln(1 - s)$$

is the binary entropy function. For small $s$, note that

$$h(s) = -s \ln s + s + O(s^2),$$

and so overall

$$\ln \binom{N}{k} = k \ln N - k \ln k + k + O(s^2 N) + O(\ln N).$$

In a regime where $s = k/N$ converges to 0, we find that the estimate $\ln \binom{N}{k} \approx k \ln(eN/k)$ is almost optimal in the sense that

$$\ln \binom{N}{k} = (k + O(1)) \ln N - k \ln k + (1 + o(1))k.$$

There is also a natural way to see this approximation from the point of view of coding theory. Consider a random subset $Y \subseteq [N]$ where each element is included independently with probability $s = k/N$. Then the entropy of $Y$ is

$$H(Y) = h(s)N = sN \ln s^{-1} + sN + O(s^2 N)$$
$$= k \ln(eN/k) + O(s^2 N),$$

the leading term of which matches our estimate for $\ln \binom{N}{k}$.

## C. Review of Chernoff Bounds

The results of Section 3.3 rely on well-known facts about tails of independent sums of "sub-Gaussian" distributions. Many references are available on this topic; for example, see Chapter 2 of (Vershynin, 2018). For completeness, here we provide an essentially self-contained proof of Proposition 2 based on the Chernoff bound for a sum of Rademacher variables.

Given a random variable $X$, define the cumulant generating function $K_X(\lambda)$ as

$$K_X(\lambda) = \ln E \exp(\lambda X).$$

For example, the cumulant generating function of a unit Gaussian $Z$ is $K_Z(\lambda) = \lambda^2/2$. *Chernoff bounds* are the following upper bounds on the probability of the tail event $X \geq a$ in terms of the cumulant generating function.

**Proposition 4.** *For $\lambda > 0$, suppose $K_X(\lambda)$ exists. Then*

$$\ln P(X \geq a) \leq -\lambda a + K_X(\lambda).$$

*Proof.* By a Markov inequality,

$$\begin{aligned}
P(X \geq a) &= P(e^{\lambda X} \geq e^{\lambda a}) \\
&\leq E \exp(\lambda X - \lambda a) \\
&= \exp(-\lambda a + K_X(\lambda)).
\end{aligned}$$

$\square$

For a unit Gaussian, this gives

$$\ln P(Z \geq a) \leq -\lambda a + \frac{1}{2}\lambda^2.$$

Minimizing with respect to $\lambda$ then gives

$$\ln P(Z \geq a) \leq -\frac{1}{2}a^2.$$

In fact, this is the best possible leading-order term; by well-known bounds on Mills ratios,

$$P(Z \geq a) = -\frac{1}{2}a^2 - \ln a + O(1).$$

Now, let $X_n$ be a sum of independent Rademacher variables, each uniformly distributed over $\{-1, 1\}$. We intuitively expect $X_n/\sqrt{n}$ to be distributed like a unit Gaussian for large $n$, and so we may hope that $P(X_n/\sqrt{n} \geq a)$ is similarly bounded as a function of $a$. A Chernoff bound lets us formalize this.

For any variable with $|X| \leq 1$, it is relatively easy to show that

$$K_X(\lambda) \leq \frac{\lambda^2}{2}.$$

For us, it is enough to know that this holds for the cumulant generating function $K_X(\lambda) = \cosh(\lambda)$ of a Rademacher variable. It follows that the same bound holds for a sum $X_n$ of $n$ independent Rademachers scaled by $1/\sqrt{n}$:

$$K_{X_n/\sqrt{n}}(\lambda) = n \cosh(\lambda/\sqrt{n}) \leq \frac{\lambda^2}{2}.$$

Therefore, for $a > 0$, we can bound the tail of $X_n$ in exactly the way that we would bound the tail of a Gaussian with standard deviation $\sqrt{n}$:

$$\ln P(X_n \geq a) = \ln P(X_n/\sqrt{n} \geq a/\sqrt{n}) \leq -\frac{a^2}{2n}.$$

This gives us the tool we need to prove Proposition 2, restated here for convenience.

**Proposition.** *Let $d > 2\epsilon^{-2}(2 \ln N + \ln p^{-1})$, and let*

$$\{F_1, \ldots, F_N\} \subseteq \{-1/\sqrt{d}, 1/\sqrt{d}\}^d$$

*be random vectors with independent, uniformly distributed entries. Then $|\langle F_i, F_j \rangle| < \epsilon$ for all $i \neq j$ with probability at least $(1 - p)$.*

*Proof.* Each inner product $I = \langle F_i, F_j \rangle$ is distributed like a sum of $d$ Rademacher variables scaled by $1/d$. By the Chernoff bound above, we have that

$$\ln P(I \geq \epsilon) = P(X_d/d \geq \epsilon) \leq -\frac{d^2\epsilon^2}{2d} = -\frac{1}{2}d\epsilon^2.$$

By symmetry $P(I \geq \epsilon) = P(I \leq -\epsilon)$, and so by a union bound

$$\ln P(|\langle F_i, F_j \rangle| \geq \epsilon) \leq \ln(2 P(I \geq \epsilon)) \leq -\frac{1}{2}d\epsilon^2 + \ln 2.$$

To conclude that $|\langle F_i, F_j \rangle| < \epsilon$ for all $\binom{N}{2} < N^2/2$ pairs of vectors with probability at least $1 - p$ by a union bound, it suffices that

$$\begin{aligned}
-\frac{1}{2}d\epsilon^2 + \ln 2 &\leq \ln \frac{p}{N^2/2} \\
&= -2\ln N + \ln 2 + \ln p,
\end{aligned}$$

which is equivalent to the condition on $d$ above. $\square$

The interested reader should also compare this result to the Johnson-Lindenstrauss lemma, which is proved in a very similar way. (See (Dasgupta & Gupta, 2003) for a proof, or the last section of (Foucart & Rauhut, 2013) for a discussion of the JL lemma with some broader context.)

## D. Review of Matched Filters

Consider the problem of inferring a scalar $S$ from the sum

$$X = Sf + Z$$

where $f \in \mathbb{R}^n$ and $Z$ is a Gaussian variable independent from $S$. Suppose for simplicity that $Z$ has non-singular covariance $\Sigma$, so that $-\ln p(z) = 1/2\|z\|_\Sigma^2$ where

$$\|z\|_\Sigma^2 = z^T \Sigma^{-1} z.$$

Then a routine calculation shows that

$$\begin{aligned}
&-\ln p(S = s | X = x) \\
&= C(x) - \ln p(s) + \frac{1}{2}\left(s - \frac{\langle f, x \rangle_\Sigma}{\|f\|_\Sigma^2}\right)^2 \|f\|_\Sigma^2 \quad (5)
\end{aligned}$$

where $C(x)$ is a constant depending only on $x$ and $\langle -, - \rangle_\Sigma$ is the inner product associated with the norm $\|-\|_\Sigma$. In particular, the distribution of $S$ conditional on $X$ is only a

function of the inner product $\langle f, X \rangle_\Sigma$. The **matched filter** for $S$ is the linear function

$$\lambda(X) = \frac{\langle f, X \rangle_\Sigma}{\|f\|_\Sigma^2},$$

and can be understood as providing the maximum likelihood estimate for $S$ conditional on $X$ under a uniform improper prior.

The quality of our matched filter is measured by its signal-to-noise ratio (SNR)

$$\rho = \frac{(\lambda(f))^2}{\operatorname{Var}_Z \lambda(Z)} = \|f\|_\Sigma^2.$$

Up to a scalar, $\lambda$ can be characterized as the linear function that maximizes this quantity. Under an improper prior, Equation (5) shows the posterior distribution on $S$ conditional on $X$ is Gaussian with mean $\lambda(X)$ and precision $\rho$.

### E. Proof of Proposition 3

We return to the proof of Proposition 3, restated here for convenience.

**Proposition.** *Let $F \in \mathbb{R}^{d \times N}$ be a Rademacher dictionary in the conditions above. Fix a $k$-element set $y \in [N]$ and some $p \in (0,1)$. If*

$$d \geq 8k(\ln N + \ln p^{-1}),$$

*then $y$ is accurately recovered from the random variable $X = Fy$ by threshold decoding with probability at least $(1-p)$.*

*Proof.* Where $X_1, X_2, \dots$ is a sequence of independent Rademacher variables of unit variance, denote

$$b(d,r) = \mathrm{P}\left(\sum_{i=1}^{d} X_i \geq \sqrt{d}r\right).$$

By a Chernoff bound, we know that

$$\ln b(d,r) \leq -\frac{1}{2}r^2 \qquad (6)$$

holds uniformly over $d$.

Now, consider a dictionary $F$ in the conditions above, and let us view its codewords $F_i$ as random vectors. Note that we can assume w.l.o.g. that $y = \{1, ..., k\}$, so that $X = Fy = F_1 + \cdots + F_k$.

Suppose that we apply threshold decoding with threshold $\tau$, so that

$$\hat{Y}_i = \begin{cases} 1 : \langle F_i, X \rangle \geq \tau \\ 0 : \text{otherwise.} \end{cases}$$

For $i = 1, \dots, k$, let $A_i$ denote the event that $y_i = 1 \neq \hat{Y}_i$. Then

$$\mathrm{P}(A_i) = \mathrm{P}(\langle F_i, X \rangle < \tau)$$

$$= \mathrm{P}\left(\sum_{\substack{j \neq i \\ j=1}}^{k} \langle F_i, F_j \rangle < \tau - 1\right).$$

The sum above is distributed like a sum of $(k-1)d$ independent Rademacher variables scaled by $1/d$. Overall,

$$\mathrm{P}(A_i) = \mathrm{P}\left(\frac{1}{d} \sum_{i=1}^{(k-1)d} X_i \geq 1 - \tau\right)$$

$$= b\left((k-1)d, (1-\tau)\sqrt{\frac{d}{k-1}}\right).$$

Similarly, for $i = k+1, \dots, N$, let $B_i$ denote the event that $y_i$ is not correctly inferred. Then the same reasoning shows

$$z\,\mathrm{P}(B_i) = \mathrm{P}(\langle F_i, F_1 + \cdots + F_k \rangle > \tau)$$

$$= \mathrm{P}\left(\frac{1}{d} \sum_{i=1}^{kd} X_i \geq \tau\right) = b\left(kd, \tau\sqrt{\frac{d}{k}}\right).$$

Overall, using Equation (6), we have

$$\mathrm{P}(A_i) \leq \exp\left(-\frac{(1-\tau)^2}{2} \cdot \frac{d}{k-1}\right)$$

$$\leq \exp\left(-\frac{(1-\tau)^2}{2} \cdot \frac{d}{k}\right)$$

and

$$\mathrm{P}(B_i) \leq \exp\left(-\frac{\tau^2}{2} \cdot \frac{d}{k}\right).$$

With $\tau = 1/2$, the probability of failure is bounded as

$$\mathrm{P}\left(\bigcup_{i=1}^{k} A_i \cup \bigcup_{i=k+1}^{N} B_i\right) \leq \sum_{i=1}^{k} \mathrm{P}(A_i) + \sum_{i=k+1}^{N} \mathrm{P}(B_i)$$

$$\leq k \exp\left(-\frac{d}{8k}\right) + (N-k) \exp\left(-\frac{d}{8k}\right)$$

$$= N \exp\left(-\frac{d}{8k}\right).$$

Setting this bound less than $p$ and rearranging proves the theorem. $\qquad\square$

### F. Possible Extensions of Proposition 3

In practice, the numerical experiments reported in Section 3.3 show that threshold decoding succeeds with little more than $d = 8k \ln N$ dimensions. In fact, it is likely possible to prove the conclusion of Proposition 3 under slightly

milder conditions by using a refinement of the Chernoff bound. For example, recall from Appendix C that the actual probability of a Gaussian tail event $Z \geq a$ is

$$\ln \mathrm{P}(Z \geq a) = -\frac{1}{2}a^2 - \ln a + O(1),$$

which is slightly less than $-1/2a^2$ for large $a$. (Note that, when $d$ satisfies the conditions of Proposition 3, the parameter $a$ used in the Chernoff bound grows on the order of $\sqrt{\ln N}$.)

Numerical experiments also showed that top-$k$ decoding succeeds with only slightly more than $4k \ln(kN)$ dimensions. We believe it is also possible to prove a bound to justify this empirical observation.

To see how, let us denote $A_{i,j}$ for the event that

$$\langle F_i, X \rangle \geq \langle F_j X \rangle.$$

Then top-$k$ decoding succeeds so long as no event $A_{i,j}$ holds for $i \in \{k+1, \ldots, N\}$ and $j \in \{1, \ldots, k\}$. Each event is identically distributed, so by a union bound we conclude that top-$k$ decoding succeeds with probability at least $(1-p)$ if

$$\ln \mathrm{P}(\langle F_{k+1}, X \rangle \geq \langle F_1, X \rangle) \leq \ln p - \ln(k(N-k)).$$

Both inner products above have variance $1/d$ and are, in some sense, approximately independent. We therefore expect that their difference can be approximated Gaussian variable with variance $2/d$. A Chernoff bound would then give

$$\ln \mathrm{P}(\langle F_{k+1}, X \rangle - \langle F_1, X \rangle \geq 0) \leq -\frac{\sqrt{d/2}^2}{2} = -\frac{d}{4}.$$

In terms of $d$, this means we need only

$$d \geq 4(\ln(k(N-k)) + \ln p^{-1})$$
$$\approx 4(\ln(kN) + \ln p^{-1}).$$

Again, we expect that improving the Chernoff bound with lower-order terms would show that only slightly more than $4k \ln(kN)$ dimensions are enough.

## G. Empirical Results on Basis Pursuit Denoising

We used the implementation of LASSO regression available in sklearn (Pedregosa et al., 2011) to infer sparse subsets of $\{1, \ldots, 2^{16}\}$ from superposition codes by minimizing the objective

$$\frac{1}{2d}\|x - F\hat{y}\|_2^2 + 10^{-5}\|\hat{y}\|_1$$

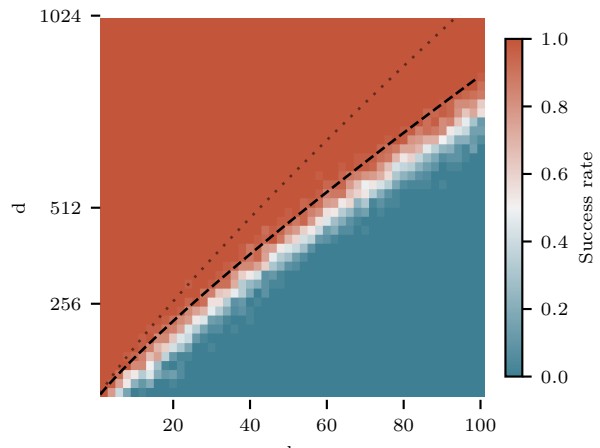

Figure 6. Empirical performance of basis pursuit decoding for $N = 2^{16}$. The bold line plots $d = k \log_2(eN/k)$, and the dotted line plot $d = 0.8k \log_2(eN/k)$.

with respect to $\hat{y}$. In compressive sensing, this is known as basis pursuit denoising (BPDN). Results are graphed in Figure 6. Compared to the performance of matching pursuit shown in Figure 5, we find that BPDN can recover a subset from even fewer dimensions; around $0.8$ bits per dimension are enough.

