# OpenReview forum: "On Bitrates of Very Sparse Superposition Codes"
_ICML.cc/2025/Conference — Submitted to ICML 2025_

### Official Review · Reviewer_576f · 2025-03-13

**Overall Recommendation:** 3

**Summary:**

This paper tackles the problem of reliability of already trained (or, hardcoded) sparse auto encoders for feature disentanglement. It does from the information theory perspective. The main result shows that 1-step methods such as SAEs are suboptimal, and that iterative methods are much more efficient. I’m very unfamiliar with the literature, so I apologise for mistakes/misunderstandings, and do not give much weight to my review.

**Claims And Evidence:**

The claims seem to be suppoerted by clear and convincing evidence.

**Essential References Not Discussed:**

I am not familiar with the literature.

**Experimental Designs Or Analyses:**

They seem on point.

**Methods And Evaluation Criteria:**

They seem to make sense.

**Other Comments Or Suggestions:**

Weaknesses: Structure of the paper is unusual: why is most of the paper Section 3? I would divide it into background/methods/results.

**Other Strengths And Weaknesses:**

The paper seems on point, as it tackles an important topic in the modern literature (SAEs), and the results seem solid.

**Questions For Authors:**

Questions: I’m aware that the model does not tackle learning, and assumes that the dictionary has been already learned, however, I feel the implications of the results in the training of SAEs or other methods should be discussed.

What could an interesting proposal be, to train a SAE, use it to extract the dictionary D, and then use the iterative method based with parameters D?

Or, would this problem be reflected in the training as well, and we should hence rely to an alternative method to discover the dictionary as well?

**Relation To Broader Scientific Literature:**

Tackles a problem in a very popular area of research.

**Theoretical Claims:**

I have checked the math only superficially, but it seems correct.

---

> ### Author Rebuttal · Authors · 2025-04-01
>
> Thanks for taking the time to write your review. We took care to make this work reasonably accessible to readers without background in compressive sensing. (See our response to Reviewer VkuK.) We're happy to see that, despite your unfamiliarity with the literature, our claims and evidence were convincing.
>
> Your question about how this work relates to training SAEs is natural. Indeed, in this work we suppose that the underlying dictionary is known and focus on the problem known as sparse reconstruction. On the other hand, an SAE also needs to learn the dictionary.
>
> The most obvious significance of our results is that the encoder layer of an SAE may not be able to reconstruct the sparse latent code with reasonable accuracy, whether or not the correct dictionary is used. In particular, it is likely that some dimensions of the latent code can _never_ be correctly inferred, and it would follow that their associated codewords can never be learned by the SAE. So, as you guessed, this suggests that we need to change our overall strategy for dictionary learning.
>
> However, we hope that our analysis—and the information-theoretic point of view in particular—can inform interpretability research beyond SAEs. We would include the main points of our response to reviewer PjJn in the conclusion of the camera-ready version.
>
> We're less sure whether the structure of our work should be adjusted. The sub-sections of Section 3 could potentially be re-numbered. However, our work doesn't follow the "methods/results" restructure common in empirical papers, and the overall sequence should remain the same.

---

### Official Review · Reviewer_XEZe · 2025-03-13

**Overall Recommendation:** 3

**Summary:**

This paper investigates the efficiency of different methods for decoding "superposition codes" - specifically focusing on the sparse reconstruction problem that appears in sparse autoencoders used for neural network interpretability. The authors focus on a simplified model where the goal is to recover a sparse binary vector (representing a k-element subset of {1,...,N}) from a lower-dimensional linear projection. They compare the performance of "one-step estimates" (simple non-iterative methods used by sparse autoencoders) with iterative methods from compressive sensing.

Key contributions:
1. Provide theoretical guarantee on the performance of one-step methods
2. Empirically show that the gap between one-step methods and iterative methods is significant

**Claims And Evidence:**

Supported claims:
- Inefficiency of one-step methods:  The theoretical guarantees in Section 3.3 and empirical results in Figure 3 convincingly demonstrate that one-step methods require approximately 2.7 dimensions per bit. The authors provide formal proofs (Proposition 3 and Corollary 2) and extensive numerical experiments across different values of N and k.
- Experiments in Figure 5 show that matching pursuit outperforms one-step methods, requiring only about 1.3 dimensions per bit. This comparison is well-documented and tested across multiple problem sizes.

**Essential References Not Discussed:**

So far so good

**Experimental Designs Or Analyses:**

The experiments make sense to me.

**Methods And Evaluation Criteria:**

Methods:
- Theoretical proof of bitrate consumption of one-step encoding methods vs. iterative methods
  - Visualization in Figure 2
- Empirical performance on synthetic data
  - Figure 3, 5

Limitations:
- Experiments are only conducted on synthetic data

**Other Comments Or Suggestions:**

N/A

**Other Strengths And Weaknesses:**

Generally the paper is clear in message and proof.

Weakness:
- The author did not discuss why efficiency is an issue in sparse autoencoders for LLM interpretation.

**Questions For Authors:**

- Can you conduct experiments on real LLM's activations and try visualize the results of one-step method vs. iterative method?
- Why efficiency is an issue in interpreting LLM activations? Usually sparse autoencoders are easy to train.

**Relation To Broader Scientific Literature:**

- Understanding superposition codes and sparse autoencoders
- Efficiency in interpreting language model activations

**Theoretical Claims:**

The proof makes sense to me.

---

> ### Author Rebuttal · Authors · 2025-03-28
>
> Thanks for taking the time to write your review.
>
> Sparse autoencoders are indeed "efficient" in the sense they are computationally inexpensive, at least relative to other dictionary learning methods. In this work, we explain that they are inefficient in an _information-theoretic_ sense. For example, we argue that, as the sparsity $k$ and latent dimension $N$ grow to infinity within a certain regime, a SAE encoder requires the dimension $d$ of a superposition code to grow faster than the entropy of the underlying sparse latent. I hope that this resolves the main weakness you perceived in this work!
>
> Your suggestion to conduct experiments using real LLM activations is natural. However, this work is occupied with providing theory to inform future methods. In our interactions with researchers doing empirical work on SAE, we realized that ideas from compressive sensing are frequently underappreciated. On the other hand, we were unable to find references characterizing the "inefficiency" of one-step methods in a regime meaningful to SAE applications. Before these insights can be used practically, we believe it's important to make the findings of the present work better known. We discuss the relevance of our work to interpretability in more depth in our response to Reviewer PjJn.

---

### Official Review · Reviewer_PjJn · 2025-03-14

**Overall Recommendation:** 2

**Summary:**

This paper investigates the efficiency of different decoding methods for superposition codes, which are used in LLM  interpretability via sparse autoencoders (SAEs). The authors compare "one-step estimates" (used by sparse autoencoders) with iterative decoding methods. There is lots of theory and some experiments. The experiments are on toy settings of sparse codes.

**Claims And Evidence:**

The paper makes clear claims about decoding efficiency that are adequately supported by both theoretical analysis and empirical results. E.g. the theoretical guarantees on one-step methods in Section 3.3 match the numerical experiments in Figure 3. However, since they only work with a toy setting the generalizability of their results cannot be fully trusted. I know of empirical evidence from real LLMs that matching pursuit methods do not improve SAEs a lot (see "Essential References Not Discussed") which makes me very concerned

**Essential References Not Discussed:**

This paper does not discuss how matching pursuits have already been tried in the SAE literature on several occasions:

* https://arxiv.org/abs/2410.14670 specifically measures encoder error with matching pursuit. It has very little impact on SAE performance on net
* https://www.lesswrong.com/posts/C5KAZQib3bzzpeyrg/full-post-progress-update-1-from-the-gdm-mech-interp-team#Replacing_SAE_Encoders_with_Inference_Time_Optimisation
* https://openreview.net/pdf?id=Pa1vr1Prww is another ITO application

To be clear, the last two references are a bit obscure. But I think references 1-2 are pretty Google-able.

**Experimental Designs Or Analyses:**

I think the experiments are generally sound. They are pretty toy and I am very skeptical about transfer to reality, however.

**Methods And Evaluation Criteria:**

The synthetic evaluation setup makes sense for studying theoretical limits of decoding efficiency. The evaluation metrics (dimensions required per bit of entropy) are appropriate for the toy setting. While the toy scenario is simplified compared to real neural networks, it's sufficient for the specific claims about coding efficiency.

**Other Comments Or Suggestions:**

No typos noticed.

**Other Strengths And Weaknesses:**

I complement the clear pedagogy of the figures of this paper.

**Questions For Authors:**

Is there some reason I should not be skeptical that your claims matter in practice, given how matching pursuits have already been tried and do not appear to boost performance that much?

**Relation To Broader Scientific Literature:**

Interpretability has wide appeal and I think if this research impacted the interp community this would have positive spillover effects

**Theoretical Claims:**

I think the claims are correct, but I am an empirical researcher not a theoretical researcher. So I cannot be confident

---

> ### Author Rebuttal · Authors · 2025-03-26
>
> Thanks for raising the question of how our findings relate to real applications of SAEs in interpretability. In fact, we hope this work will help interpretability researchers become more aware of ideas from compressive sensing. However, after reading your review, we realize it may be useful to clarify the broader significance of these ideas.
>
> Let's begin by addressing your main concern. It's true that attempts to apply iterative reconstruction at inference time, as in Engels' work, have not had much success. However, the dictionaries used by these inference-time optimization methods are still learned using non-iterative ("one-step") encoders at training time. If some latent variable cannot be reliably inferred during training, its codeword will almost certainly not become an element of the dictionary, and no amount of optimization at inference time will be able to read the latent. So, the failure of inference-time optimization (ITO) does not rule out the possibility that SAE error could be modeled by an iterative encoder provided with the right dictionary. Indeed, when its bitrate exceeds the capabilities of a top-$k$ encoder, a superposition code from our own toy example will look like SAE dark matter in the language of Olah/Engels.
>
> In our own work, however, we are explicitly holding off on proposing an improvement over SAEs. Our message is _not_ that SAEs need to be modified to use specific compressive sensing algorithms, or that real experiments with residual vectors will resemble our own toy experiment. Our message is simply that, in general, reading latent variables from a superposition code through "one-step estimates" is very inefficient from a coding-theoretic point of view and that even surprisingly naive methods (like matching pursuit) can do much better. This is true for generic reasons and is a well-known phenomenon in the field of compressive sensing. However, before this work, we weren't able to find a comparative analysis of SAE-type encoders in a regime that makes sense to interpretability researchers.
>
> How can this message inform work on SAEs? As we see it, one huge challenge in mechanistic interpretability is that we don't know the true structure of the (hypothetical) interpretable latent representation. For example, it's possible that some "features" have special relationships, should be viewed as elements of a continuous space, etc.. However, we can still talk about the entropy of the latent representations and consider the bitrate at which it is encoded by an activation vector.
>
> Even beyond the toy model of this paper, it is very natural that linear estimates for properties of the latent representation would suffer from significant crosstalk. As we explain in this work, this limits one-step methods to reading a small amount of information—for example, a fractional bit—from each dimension of the residual vector. (In practice, it's possible that only a limited selection of "large magnitude" properties will be recovered, while latent variables represented with smaller magnitude are impossible to read due to crosstalk.) The compressive sensing point of view, as introduced in this paper using a simple toy example, raises at least two routes for future inquiry:
>
> 1. First, the _existence_ of compressive sensing algorithms means that significantly more information can be stored in a residual vector through linear superposition. Although neural networks do not explicitly decode the latent representation—let alone run a method like OMP, ISTA, etc.—there is no reason in principle why they cannot learn to use this extra information, especially if the latent signal has some special structure. This raises the question of estimating the bitrates of neural representations in practice and determining whether they exceed the low bitrates that limit one-step decoders.
> 2. The structure of existing compressive sensing algorithms gives us some clues on how this "extra information" can be decoded, should it exist. Roughly, the idea is to use our dictionary to model the noise term $Z$ of Equation 1 (line 216), which previously we treated as a Gaussian. In practice, this means letting our estimates for the different components of the latent vector communicate.
>
> Overall, the hypothesis that extra information can be stored in linear superposition is _not ruled out_ by the apparent failure of ITO, and we believe the point of view of compressive sensing can help us reason about future approaches in interpretability.
>
> If this resolved your skepticism regarding the failures of ITO, please consider reevaluating our work. If you found this explanation helpful, we would be happy to include some similar discussion in the conclusion of the camera-ready version.

---

> > ### Comment · Reviewer_PjJn · 2025-04-04
> >
> > > the dictionaries used by these inference-time optimization methods are still learned using non-iterative ("one-step") encoders at training time
> >
> > I am not very compelled by this reasoning. Firstly, SAE dictionary sizes are generally many times greater than the base dimension. I would expect that if matching pursuit-style methods, such as the one used here: https://www.alignmentforum.org/posts/C5KAZQib3bzzpeyrg/full-post-progress-update-1-from-the-gdm-mech-interp-team#Replacing_SAE_Encoders_with_Inference_Time_Optimisation were a very promising approach in LLMs then due to the large size of the dictionaries used here then this would lead to large performance improvements in reality. But there is a very marginal improvement.
> >
> > Additionally, when you discuss details such as
> >
> > > How “efficient,” in terms of bitrate, are the codes used by real neural networks...
> >
> > This is closely related to existing discussion here: https://transformer-circuits.pub/2023/may-update/index.html#dictionary-worries on whether compressed sensing is "too strong". Simple one-layer MLPs as the architecture of choice for SAEs is not totally random, it is based on the observation that neural networks (particulalry transformers) are highly linear (the linear representation hypothesis, https://arxiv.org/abs/2311.03658).
> >
> > You could argue that more powerful compressed sensing algorithms are underexplored by the LLM community. This is quite possible to me, hence why I am not strong rejecting this paper. However, without any validation of the interpretability, naturalness (is a transformer really using this representation vs my algorithm using it) and performance of this technique on any real neural network, I wish to mantain my score

---

> > > ### Author Response · Authors · 2025-04-07
> > >
> > > You are right that SAE dictionary sizes should be much larger than the base dimension. In our own experiments, we considered codeword dimensions of around $d=1024$ and dictionary size of up to $N = 2^{20} = 1048576,$ and also argued that similar results would hold in general when $N$ is _exponentially larger_ than $d.$
> > >
> > > However, you also suggest that a better inference method should be able to provide a better model for activation vectors merely because the dictionary learned by an SAE is large:
> > >
> > > > I would expect that if matching pursuit-style methods [...] were a very promising approach in LLMs then due to the large size of the dictionaries used here then this would lead to large performance improvements in reality.
> > >
> > > It's certainly not true that a method capable of decoding sparse latents when provided with the correct dictionary should also display "performance improvements" over a baseline when provided with an incorrect or incomplete dictionary, even when the dictionary in question is very large!
> > >
> > > In the problem setting from our own paper, consider whether a superposition code $y = F x$ can be "decoded" in any sense by applying matching pursuit over a different, independent dictionary $F'.$ Even if $F'$ has an exponentially large number of codewords, Proposition 2 (Section 3.3) tells us that, with high probability, all of these vectors have relatively small cosine similarity with the codewords belonging to $y$ [1]. So, it is certainly not true that elements of $F'$ can be expected to play the role of true codewords. An iterative method like orthogonal matching pursuit can _trivially_ minimize the reconstruction loss of $y \approx F' x',$ but only at the expense of making $x'$ much less sparse than $x$ [2]. Overall, it's clear that sparse inference in this setting is simply not possible in any meaningful sense, and improving "performance" on this problem is neither necessary nor sufficient for an inference method to succeed in a setting where the true dictionary is known.
> > >
> > > It follows that the hypothesis that the "dark matter" of an SAE could be decoded by matching pursuit _is consistent_ with the failure of ITO (inference-time optimization), so long as the true dictionary atoms are not learned by the SAE in its training phase. This work also indicates why an SAE may be incapable of learning these atoms: specifically, we showed why sparse latents that a simple algorithm can infer may nevertheless have optimal linear estimates (matched filters) with vanishingly small signal-to-noise ratio.
> > >
> > > We agree that our question on bitrates in real neural networks is closely related to Henighan and Olah's question of whether compressive sensing is "too strong." However, we would highlight that the information-theoretic point of view is essentially novel; as far as we know, the "information efficiency" of SAEs was not studied prior to this work. Therefore, our work provides a new vantage point on this question.
> > >
> > > Overall, you argued our paper might not be relevant because the failure of ITO experiments proves that matching pursuit would not be capable of decoding activation vectors in practice, even if we knew the true dictionary. However:
> > >
> > > 1) Your reasoning seems to be at odds with fundamental ideas described in our paper, as detailed above.
> > > 2) As we explained in our original rebuttal, our work gives insight into the generic problem of sparse inference in the regime encountered by SAEs, and our conclusions are meaningful independent of any specific compressive sensing method. The value of our work is _not_ conditional, e.g., on matching pursuit resulting in performance improvements.
> > > 3) Your suggestion that stronger compressive sensing algorithms may perform better at ITO means that our findings _are_ relevant to practitioners since, in this case, it would be important to benchmark the performance of different algorithms on a well-understood problem. Again, we highlight that compressive sensing is classically studied in a "linear regime" where the undersampling ratio $N/d \approx \rho$ is bounded above by a moderate constant. Also note that we included results on $L_1$ coordinate descent in Appendix G, and will refer to these results in the main text of the camera-ready version.
> > >
> > > [1] For example: Proposition 2 implies that, for a dictionary of size $N = 2^{20},$ it's true with probability at least $p = 1/2$ that no element of $F'$ has absolute cosine similarity of more than $1/4$ with any fixed element of $F$ so long as $d > 2 \times (1/4)^{-2}  \times (2 \ln(2^{20}) + \ln 2) > 909.$ In practice, such a large absolute cosine similarity turns out to be very unlikely even when $d = 512.$
> > >
> > > [2] For example, we found empirically that when $N = 2^{20}$ and $d = 512,$ we need around $30$ steps of orthogonal matching pursuit on the dictionary $F'$ to model about 80% of the squared norm of a single codeword drawn from the unknown dictionary $F.$

---

### Official Review · Reviewer_VkuK · 2025-03-14

**Overall Recommendation:** 2

**Summary:**

This paper investigates the efficiency of one-step estimates used in sparse autoencoders for recovering latent representations from neural network activations. The key contribution is an analysis of the bitrate required for reliable decoding. The study demonstrates that one-step estimates require significantly more dimensions per bit compared to iterative sparse recovery methods. This highlights a fundamental inefficiency in current autoencoder-based interpretability techniques and raises the question of whether neural networks encode information more efficiently than simple one-step decoders can extract.

**Claims And Evidence:**

The main claim of the paper, i.e., one-step code estimates require more dimensions per bit than matching pursuit, seem to be reasonably demonstrated.

**Essential References Not Discussed:**

As mentioned, I believe there is a serious lack of relevant references. While the authors discuss compressive sensing and some of Candès’s work, the most relevant (RIP, Candès, 2008) is missing. Moreover there is no discussion of proximal algorithms, whose unrolled versions have been heavily studied (Gregor and LeCun, 2010) and would have been a more natural iterative algorithm to use.

**Experimental Designs Or Analyses:**

The experimental results seem convincing, but are limited both in breadth but also as mentioned in the methods section.

**Methods And Evaluation Criteria:**

While reasonable tools are used in order to investigate the questions, I’m surprised that despite being a relatively thorough discussion of compressive sensing and coding algorithms, there was no mention of coherence/RIP (which clearly, theoretically, explain why random kernels do optimally well) nor any discussion of proximal algorithms and unrolled architectures (which would be better iterative methods than OMP).

**Other Comments Or Suggestions:**

No further comments, everything was addressed in the previous sections.

**Other Strengths And Weaknesses:**

Figure 1 is a nice visual and conveys an important concept cleanly. The discussion on 222 of $\epsilon$-orthogonality feels redundant since there are multiple concepts (coherence, mutual coherence, RIP) to express this in a more principled way.

**Questions For Authors:**

No further questions, everything was addressed in the previous sections.

**Relation To Broader Scientific Literature:**

The work is very relevant to the scientific literature: sparse coding has a long history of drawing researcher’s attention, and it is particularly popular currently to explain large models.

**Theoretical Claims:**

I did read all the theoretical sections, and while I did not spot any obvious errors, I did not examine them in depth.

---

> ### Author Rebuttal · Authors · 2025-03-26
>
> Thank you for your thoughtful feedback and for bringing up the connection with compressive sensing (CS). Our decision to not reference many tools from this world—like restricted nullspace properties, proximal gradient methods, etc.—may be surprising, but it was made deliberately.
>
> During this work, we did our best to inform ourselves in CS and were impressed by the variety of tools and perspectives. For example, besides the RIP, the restricted nullspace property also guarantees recovery of a sparse vector from the $L_1$ minimization problem but has been found to hold under weaker conditions [1]. On the side of numerical methods, many iterative algorithms (like ISTA and FISTA) can be motivated using proximal gradient descent, but Dono, Maleki, and Montanari showed that belief propagation motivates a proximal-type iteration with an additional "correction," sometimes called the Onsager correction, that significantly decreases the required undersampling ratio compared to ISTA [2]. It's hard to do this field justice! We appreciate your suggestion to reference another work of Candès'—are you referring to [3]?
>
> However, we believe that the main content of this work wouldn't be improved by more tools from CS. This is for two reasons:
>
> 1. Our main message can be stated and defended without relying on tools like the RIP or reconstruction algorithms beyond simple matching pursuit.
> 2. As far as we know, the findings of this paper—in particular, the empirical performance of matching pursuit in Section 3.5—cannot be easily explained by existing theory of compressive sensing.
>
> To begin, let us address your concern with our use of $\epsilon$-orthogonality/incoherence in Section 3.3. You suggest that it would be better to rely e.g. on the RIP. In fact, this section does not deal with the standard framework of compressive sensing: its goal is only to understand the reliability of what we call _one-step estimates_. (Such an estimate is like the first iteration of an iterative method.) As we explain, what determines the success of a one-step estimate is the scale of the crosstalk between different code words. Our proof of Proposition 3 (the main result of the section) relies on a uniform bound over sums of crosstalks. In usual regimes of compressive sensing, where the undersampling and sparsity ratios are bounded below by positive constants, these sums become very large and it becomes easy to show that one-step estimates do not work. Therefore, the results of Section 3.3 and the "rules of thumb" shown in Figure 3 have little to do with compressive sensing and would not benefit from a discussion of the RIP or the RNP.
>
> You also suggested that it would be more natural to use a proximal gradient method. In fact, we previously considered using ISTA as our sparse reconstruction method. In Appendix G, we show some results obtained using coordinate gradient descent for the $L_1$ objective, which is a common alternative to ISTA that turned out to work better in our experiments. (We will mention this appendix in the main text of the camera-ready version.) ISTA performs slightly better than matching pursuit, but not enough to contribute to our conclusions substantially. Indeed, the role of Section 3.5 is only to show that _some_ simple iterative method can significantly outperform one-step methods in our chosen regime, and our observation that one of the simplest imaginable methods already performs well strengthens this conclusion. The problem of benchmarking different compressive sensing methods is out of scope.
>
> Finally, even if we had introduced more compressive sensing theory, we would not have been able to provide significantly more insight about the empirical results of Section 3.5. A lot of classical CS theory, like [4], focuses on the "linear regime" where $k/d \to \rho$ and $d/N \to \delta$ for some moderate constants $(\rho, \delta)$ bounded strictly between $0$ and $1,$ and applying bounds from this theory (like bounds derived from the RIP) give very weak guarantees in our setting. We searched the literature for more information on sublinear regimes where $\ln k / \ln N \to \epsilon < 1,$ including works like [5], but were ultimately not able to find a practical guarantee that explained Figure 5, and decided that further inquiry was out of the scope of this work.
>
> Overall, we hope that this work encourages some readers to learn more about CS and are happy to include some more pointers and references in Section 3.5. (Certainly, it was our own inspiration.) However, including more CS theory in the body of the work would not make our findings significantly easier to explain or to understand. Given the choice, we prioritized making this work as accessible as possible.
>
> [1]  https://doi.org/10.1016/j.laa.2016.03.022
> [2]  https://doi.org/10.1073/pnas.0909892106
> [3]  https://doi.org/10.1016/j.crma.2008.03.014
> [4]  https://doi.org/10.1109/TIT.2005.858979
> [5]  https://proceedings.mlr.press/v99/reeves19a.html

---

### Decision · Program_Chairs · 2025-05-01

**Decision:**

Reject

**Comment:**

This work studies the efficiency of "linear superposition codes" with the goal of understanding how large scale models embed information into latent states. The work is primarily theoretical and aims to understand the difference between direct estimation and iterative solvers in decoding these superposition codes. The reviewers generally appreciated the approach, however it was clear from the reviews and the discussion that the theory lives in the area of compressive recovery, and a deeper discussion and understanding on how this theory connects to the large existing literature. Moreover one reviewer had identified a challenge in the scaling of the results up to real-world sizes that can be quite challenging in both theory and practice. Thus while it seems that there is some nice foundations in this work, a bit more understanding of the extent to which the work complements the current literature and can address problems of interest is vital.